# Effects of Annealing on Surface Residual Impurities and Intrinsic Defects of $\beta$-Ga$_2$O$_3$

**Songhao Wu**, **Zichun Liu, Han Yang and Yeliang Wang** *

MIIT Key Laboratory for Low-Dimensional Quantum Structure and Devices, School of Integrated Circuits and Electronics, Beijing Institute of Technology, Beijing 100081, China; songhao.wu@bit.edu.cn (S.W.)
* Correspondence: yeliang.wang@bit.edu.cn

**Abstract:** In this study, the effects of annealing on the surface residual impurities and intrinsic defects of unintentionally doped (UID) $\beta$-Ga$_2$O$_3$ are investigated by adopting high-temperature thermal treatments at 1000 °C for 1 h under vacuum and O$_2$ ambience. It is found that the recovery between the divacancies V$_{Ga}$+V$_O$ and interstitials (O$_i$) occurs during annealing, and the residual impurities are identified as Si and Cr, which are repelled toward the surface during annealing. Interestingly, these impurities occupy the formation of Ga vacancies (V$_{Ga}$) near the surface formed by oxygen annealing, consequently weakening the relevant impurity scattering and improving carrier mobility. Moreover, the carrier density of the samples is explored using temperature-dependent Hall measurements, which show a slight reduction in both vacuum and oxygen annealing. This reduction might be a result of the V$_{Ga}$ pushing the Fermi level away from the conduction band. In addition, the activation energy of Si ions occupying V$_{Ga(I)}$ is lower than that of the interstitial Si ions.

**Keywords:** $\beta$-Ga$_2$O$_3$; annealing; defects; activation energy

## 1. Introduction

As an emerging ultra-wide bandgap semiconductor, beta gallium oxide ($\beta$-Ga$_2$O$_3$) has attracted extensive research attention due to its specific properties. From an optoelectronic perspective, $\beta$-Ga$_2$O$_3$ shows great potentiality in the deep-ultraviolet range [1]. $\beta$-Ga$_2$O$_3$ shows a large bandgap (4.5–4.9 eV) and high breakdown electric field (~8 MV/cm) to produce a Baliga's figure of merit (BFOM) over 3000, which is much higher than SiC BFOM and GaN BFOM. As a result, $\beta$-Ga$_2$O$_3$ is superior in power device applications, the performance of which can be conveniently modulated due to its good n-type doping controllability over a wide range of 10$^{15}$–10$^{19}$ cm$^{-3}$ [2–5]. Based on its ultra-wide bandgap, $\beta$-Ga$_2$O$_3$ also demonstrates the exclusive potential for use in optoelectronic detection for solar-blind wavelength (<280 nm), which matches its bandgap well. For harsh environment operations, inherent stability is considered a crucial property.

Although large-area bulk Ga$_2$O$_3$ materials are becoming available due to the significant breakthrough in growth technology [6], the crystal quality requires further improvement, as it is limited by point defects [7,8]. Generally, point defects in bulk Ga$_2$O$_3$ are classified as native defects and extrinsic defects, which originate from host atoms and impurity atoms. Two nonequivalent gallium (Ga) atoms exist in Ga$_2$O$_3$, which are labeled as Ga(I) and Ga(II) [9]. With three valence electrons, a missing Ga atom would leave three oxygen-dangling bonds, indicating that a gallium vacancy (V$_{Ga}$) can act as a triple acceptor-like native defect [10]. Quoc et al. have studied the V$_{Ga}$-related energy levels in the bandgap, suggesting that stable transition energy is 1.4 eV below the conduction band for V$_{Ga(I)}^{3-}$ and 2.0 eV for V$_{Ga(II)}^{3-}$ [11]. Moreover, oxygen atoms have three different sites referred to as O(I), O(II), and O(III), which can correspondingly produce deficient oxygen vacancies (V$_O$) with energy levels of 1.6 eV, 2.0 eV, and 1.7 eV below the conduction band, respectively [11]. However, these V$_O$ barely contribute to n-type conductivity due to their deep energy levels.

Other origins of point defects are usually oxygen interstitials ($O_i$) in bulk crystals, which are, to some extent, particularly neglected regarding the thermal dynamics and their effects on the material properties.

On the other hand, extrinsic defects are always induced by residual impurities like Si and Cr, which originate from the source material ($Ga_2O_3$ powder). By substituting Ga sites in $Ga_2O_3$, silicon impurities usually behave as shallow donors with activation energies ranging from 10 to 50 meV [12–14]. As previously reported, two types of substitutional Ga sites by Si, namely, $Si_{Ga(I)}$ and $Si_{Ga(II)}$, can be efficiently formed, and Si impurities are reported to preferentially occupy the Ga(I) sites [15]. The interstitial Si ($Si_i$) is a shallow donor at 150 meV and is stable in the charging state q = +1 [16,17]. In addition, Cr impurities from the ambience during bulk material growth can be incorporated into the $Ga_2O_3$ lattice by filling these Ga sites to form $Cr^{2+}$ and $Cr^{3+}$ states [18], and the former charge state is always dominant in the as-grown $Ga_2O_3$ [19].

The aim of this study is to investigate the effect of annealing on diverse defects and behaviors in $Ga_2O_3$, among which the defects related to oxygen vacancies are the most common in metal–oxide materials. As a result, simple annealing in oxygen is chosen in this study, and the vacuum ambience is also selected for comparison to reflect the effects of oxygen, which is accompanied by another reference sample annealed in both ambiences. The secondary-ion mass spectroscopy (SIMS) is used to identify residual impurities such as Si and Cr, which are demonstrated to be repelled toward the $β$-$Ga_2O_3$ surface by annealing. Moreover, the behaviors of intrinsic defects are also studied by Raman and PL spectroscopy to further understand the underlying effects of annealing on $Ga_2O_3$. Furthermore, the effects of residual Cr and Si impurities on the electrical properties of bulk $Ga_2O_3$ are also explored via temperature-dependent Hall measurements.

## 2. Experiment

Bulk crystals of UID (-201) $β$-$Ga_2O_3$ with a thickness of about 480 μm were purchased from Beijing MIG Semiconductor Co., Ltd., which were grown by the edge-defined film-fed (EFG) method. They were diced into 1 cm × 1 cm pieces, and the substrates were cleaned ultrasonically with acetone, ethanol, and de-ionized water for 10 min, respectively. Then, the substrates were annealed in vacuum and oxygen ambience, during which the annealing temperature was held at 1000 °C for 1 h with a heating rate of around 16.2 °C/min, and cooled down to room temperature. The vacuum-annealed sample was processed in an evacuated chamber at a pressure lower than 2 Pa using a mechanical pump, whereas the $O_2$-annealed sample was processed in a 25 sccm $O_2$ ambience (33 Pa). An additional pristine UID $β$-$Ga_2O_3$ sample without annealing was also prepared as the control sample.

The annealing effects on the crystal quality of $β$-$Ga_2O_3$ were systematically investigated using X-ray diffraction (XRD), Raman spectroscopy, photoluminescence (PL) spectroscopy, and secondary-ion mass spectrometry (SIMS). Their XRD was carried out to investigate the crystalline quality of the samples at a scan rate of 0.02 degrees. The Raman spectra were recorded using a front configuration setup with a 532 nm green laser as the excitation source and were calibrated using the 521.0 $cm^{-1}$ peak of silicon at room temperature (25 °C). The diameter of the laser was about ~20 μm (objective: 50×), and the power was 150 mW with a 5 s integral time, focusing on the surface. The PL spectra of the samples were collected using a 74× objective excited by a 325 nm CW HeCd laser with 10 s integral time. The relative positions of all samples tested were similar to ensure the accuracy of the data as much as possible.

In addition, temperature-dependent Hall measurements were conducted from 200 to 400 K to explore the impact of annealing on the carrier concentration and mobility of the samples. Ti/Au (20/200 nm) were deposited at the four corners of the $Ga_2O_3$ samples and annealed in $N_2$ for 5 min to obtain good ohmic contacts for Hall measurements, and the measured resistance between the electrodes was acceptable at lower than 10 Ω.

## 3. Results and Discussion

As shown in Figure 1a, three sharp XRD peaks appear for the diffraction on the (-201), ($\bar{4}$02), and ($\bar{6}$03) planes of $\beta$-Ga$_2$O$_3$. For further comparison, the ($\bar{2}$01) diffraction peak and the full width at half maximum (FWHM) were selected and investigated, as shown in Figure 1b,c. According to the intensity of the (-201) plane, the computed texture coefficients for the unannealed, vac-annealed, and O$_2$-annealed samples were 0.91, 0.87, and 0.76 (detailed calculations are provided in the supplementary information). Each sample was measured several times to minimize possible fluctuations. The average FWHM of the ($\bar{2}$01) diffraction peak was 44.2 arcsec for the pristine sample, which reduced to the average value of 38.3 arcsec due to vacuum annealing. The reduced FWHM indicates an improvement in bulk crystallinity, and the resultant dislocation density decreased from $3.1 \times 10^4$ to $2.3 \times 10^4$ cm$^{-3}$ as extracted from the FWHMs [20,21]. However, although the FWHM of the O$_2$-annealed sample (44.6 arcsec) was similar to that of the control sample, an obvious shoulder peak appeared (Figure 1b: red line), which was attributed to the strain accumulation induced by the diffusion of Si and Cr impurities toward the surface (detailed discussion provided in the SIMS section).

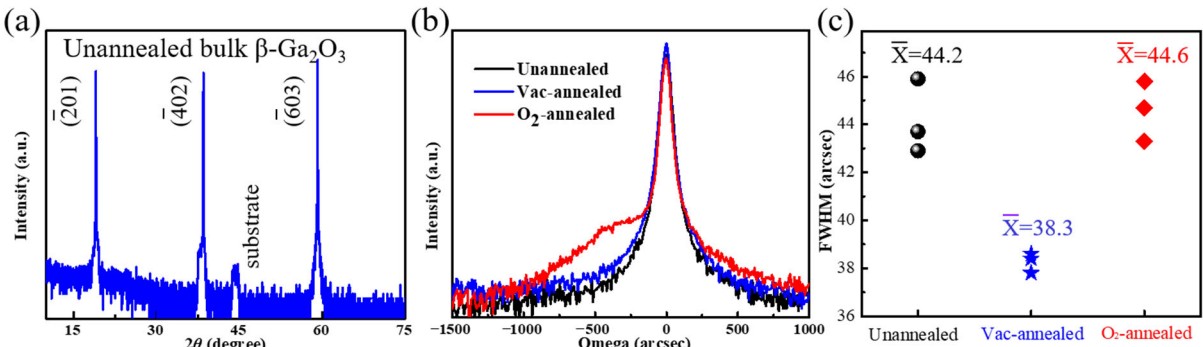

**Figure 1.** XRD measurement results for the three types of Ga$_2$O$_3$. (**a**) The 2-theta scan of the unannealed Ga$_2$O$_3$. (**b,c**) The ($\bar{2}$01) plane rocking curves of all samples.

The Raman spectra of the samples are presented in Figure 2a [22]. According to the calculation of the factor group analysis at the $\Gamma$ point [23], 27 optical phonon modes exist as $\Gamma_{opt} = 10A_g + 5B_g + 4A_u + 8B_u$, among which A$_g$ and B$_g$ are Raman active, whereas A$_u$ and B$_u$ are infrared active [24]. Although the Raman mode intensity can be affected by the crystal orientation and surface content of the vibrating group [23], the information on the latter is believed to be dominantly reflected by the Raman measurements because the crystal orientation and test conditions of the samples are completely consistent. All peaks are normalized via calibration based on the Si substrate. Compared to the pristine sample, the Raman peaks that vary predominantly for the annealed samples are listed in Figure 2b. Although the B$_g$(5) mode is believed to be dependent only on Ga(I) and O(III) [25], its weakness for the O$_2$-annealed sample implies the reduction in Ga(I) instead of in O(III) because the concentration of O(III) is difficult to drop in an oxygen-rich ambience due to the large energy formation (E$_f$) of V$_{O(III)}$ around 4.7 eV [11]. As a result, the weakened B$_g$(5) strongly suggests an increase in Ga-related vacancies with reduced Ga(I) concentration. On the other hand, the strengthened B$_g$(5) mode of the vacuum-annealed sample demonstrates an increased O(III) concentration, which results from the recovery with O$_i$ point defects, as discussed later. Additionally, Raman modes other than the B$_g$(5) mode are related to at least three types of atoms [25], which makes it difficult to provide solid evidence.

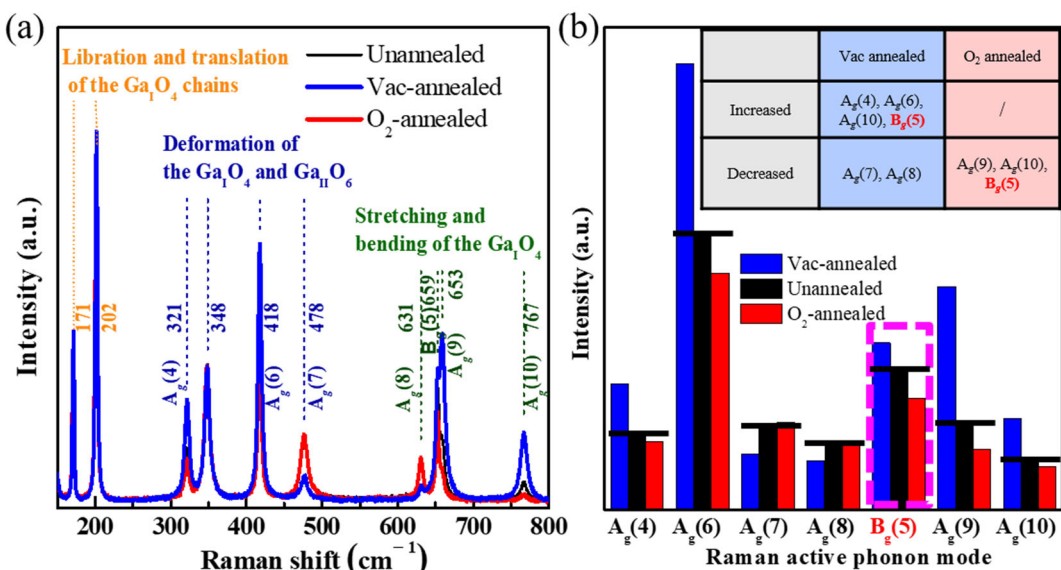

**Figure 2.** (**a**) Raman shift of three types of $Ga_2O_3$ from 150 to 800 cm$^{-1}$. (**b**) The peaks varying predominantly for annealed samples are listed.

The SIMS measurement is used and corrected to identify the residual impurity compositions in the pristine, vac-annealed, and O$_2$-annealed Ga$_2$O$_3$, demonstrating the existence of Si and Cr impurities, the former of which is the main residual impurity, as shown in Figure 3a. According to the SIMS measurement, the normalized concentrations of impurities are extracted from all the samples, in which the relative values of Cr$^+$, Si$^+$, and Ga$^+$ are 1/176/385,987, 1/80/2936, and 1/303/31,665 for the unannealed, O2-annealed, and vac-annealed samples, respectively. For the vac-annealed sample (Figure 3b), the Ga concentration near the surface is almost unchanged, but the Si and Cr concentrations are higher than that of the pristine sample, which indicates that annealing can promote the residual Si and Cr out-diffusion toward the surface. In Figure 3c, it can be seen that the Ga concentration near the surface of the O$_2$-annealed sample is lower than that of the pristine and vac-annealed samples, which indicates the formation of V$_{Ga}$ near the surface caused by O$_2$ annealing [26]. In addition, the increasing content of Si and Cr impurities near the surface of the O$_2$-annealed sample strongly indicates the migration of Si and Cr impurities toward the surface by O$_2$ annealing, which can occupy the V$_{Ga}$ to promote the formations of Si$_{Ga}$ and Cr$_{Ga}$. The migration of Si and Cr impurities toward the surface leads to the accumulation of strain [26], which induces a shoulder peak in the corresponding XRD pattern (Figure 1b). The SEM images of the surface morphologies of all three samples are shown in the SI. Figure 2 shows the degraded surface of the O$_2$-annealed sample.

Figure 4a presents the room-temperature PL spectra of the pristine, oxygen-annealed, vacuum-annealed, and (vacuum+oxygen)-annealed samples. The spectral peaks centered at 408 nm (3.04 eV) and 508 nm (2.44 eV) can be observed for all samples, although their intensities significantly subside for both vacuum and oxygen annealing. According to the calculation by Quoc et al., the defects of V$_{Ga}$+V$_O$ and O$_i$ are located at ~3.0 eV and ~2.4 eV below the conduction band minimum (CBM) to act as deep acceptors (detailed calculated defect levels are shown in Supplementary Figure S1) [11,27]. During the PL measurement, electrons trapped by these acceptors rather than in the valence band are believed to be excited in the conduction band because the 325 nm excitation photos have lower energy than the bandgap of Ga$_2$O$_3$. These excited electrons transit to the near-CBM energy levels of the Si shallow donor via non-irradiative processes and subsequently drop to the deep acceptors to generate violet and green luminescence, which are related to the defects of V$_{Ga}$+V$_O$ and O$_i$, respectively [11]. Therefore, the weakened violet and green PL peaks for the annealed samples reflect the reduction in both defects, which result from the enhanced combination of intrinsic O$_i$ and the divacancy V$_{Ga}$+V$_O$ by absorbing thermal energy during

annealing. Notably, the green PL peak is less weakened for the oxygen-annealed sample than for the vacuum-annealed sample because ambient oxygen (25 sccm) can be absorbed to compensate for $V_{Ga}+V_O$ so that fewer intrinsic $O_i$ is consumed.

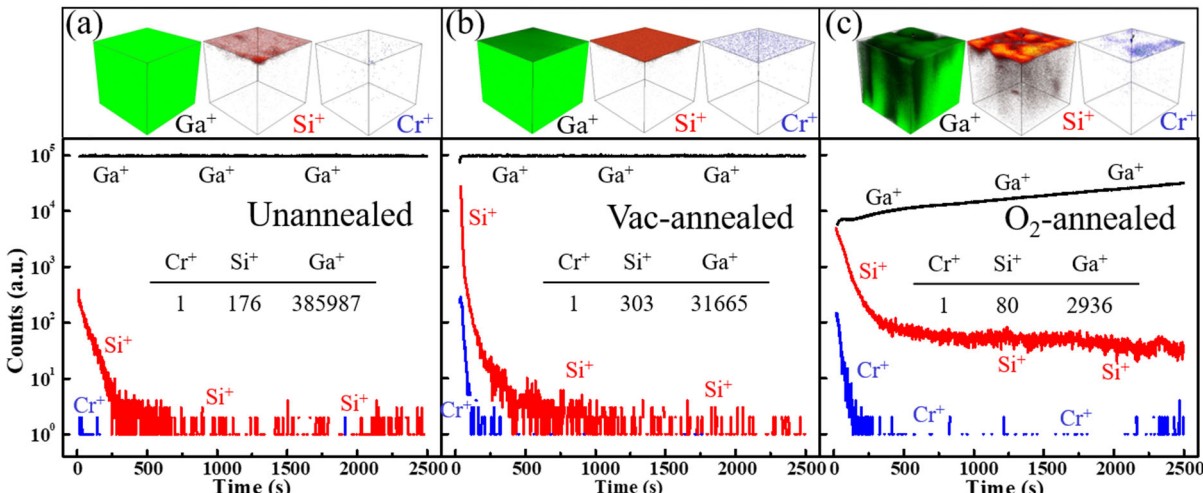

**Figure 3.** The in-depth SIMS profile of Ga, Si, and Cr for the (**a**) unannealed, (**b**) vac-annealed, and (**c**) $O_2$-annealed $Ga_2O_3$.

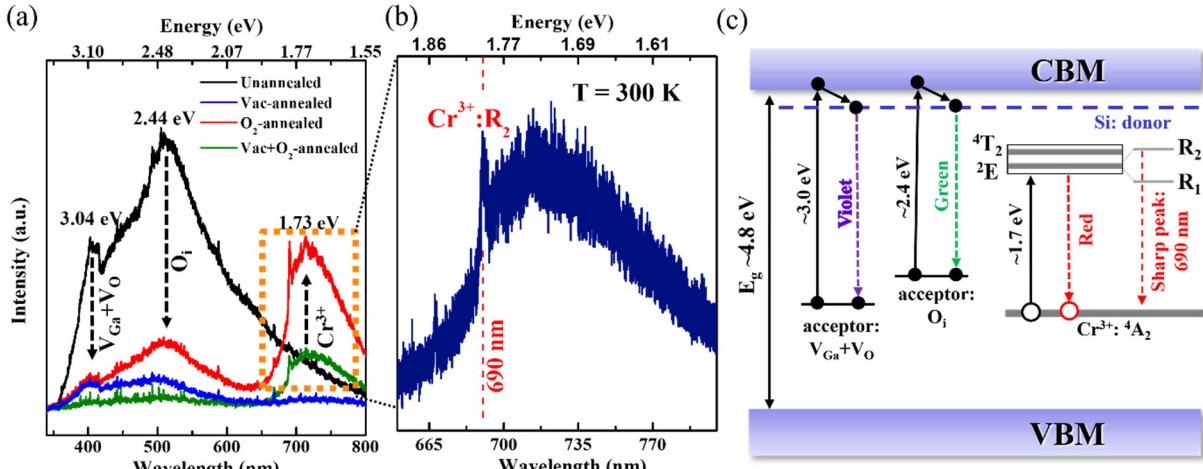

**Figure 4.** (**a**) PL spectra at an arbitrary scale for the four crystal types. (**b**) Magnified red PL spectrum of $Ga_2O_3$ annealed in $O_2$ from 650 nm to 800 nm. (**c**) PL mechanism in $Ga_2O_3$: violet, green, and red emissions.

Interestingly, a broadband red PL peak centered at 716 nm is observed only for the sample annealed in $O_2$, which is associated with a spike peak (690 nm) on its shoulder (Figure 4b). An additional pristine sample is sequentially annealed in a vacuum and oxygen ambience to verify the existence of the red PL peak, which also appears (Figure 4a: green line). Although the defects of $V^0_{O(III)}$ are located at 1.7 eV below the conduction band, which corresponds to a wavelength of around 716 nm [11], it should not be the origin of the red PL peak because $V_O$ is difficult to form in oxygen ambience due to its high formation energy of 4.7 eV [10,11]. Furthermore, the broad red PL peak can be attributed to the electron transitions from the $^4T_2$ and $^2E$ states to the $^4A_2$ levels of the $Cr^{3+}$ impurities [28–32], which are generated by recharging $Cr^{2+}$ during thermal annealing to shift the Fermi level away from the conduction band [21,22]. As for the spike peak at 690 nm, the transition of the $^2E:R_2$ states to ground $^4A_2$ is believed to be the dominant origin [18,33,34].

Temperature-dependent Hall measurements were conducted to further explore the annealing effects. As shown in Figure 5a, the electron mobility of the three samples in

this study is comparable at 300 K, which is 100 cm$^2$/V·s, and is in good agreement with previous studies [10,35]. The carrier mobility was mainly determined using lattice vibration scattering and ionized impurity scattering, which can be expressed as exponents $T^{-1.5}$ and $T^{1.5}$, respectively [36]. The closest exponent to the theoretical $T^{-1.5}$ for the vacuum-annealed sample suggests its improved crystallinity to enhance lattice vibration scattering, as supported by the reduction in both $O_i$ and $V_{Ga}+V_O$ defects (the fitting process is shown in Supplementary Table S1).

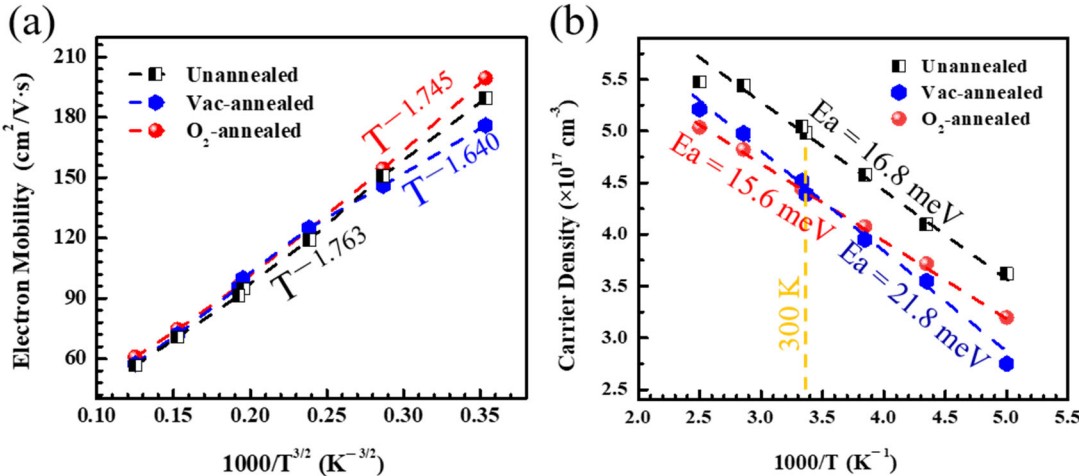

**Figure 5.** The temperature-dependent (**a**) electron mobility and (**b**) carrier density for the three types of bulk $Ga_2O_3$.

At 200 K ($1000/T^{3/2} \approx 0.35$), the oxygen-annealed sample shows higher mobility than the pristine $Ga_2O_3$, whereas the vacuum-annealed sample is lower because annealing in oxygen increases the amount of defective $V_{Ga}$ to provide more sites for Si and Cr impurities. Consequently, substitutions of $Si_{Ga}$ and $Cr_{Ga}$ are formed, thus weakening the impurity scattering for carrier mobility improvement. On the other hand, annealing in a vacuum repels Si and Cr from the lattice to enhance ionized impurity scattering, which is dominant at this temperature [33].

The carrier concentration and activation energy of Si-doped $Ga_2O_3$ are listed in Table 1. Our results are in good agreement with those of the previous studies. As shown in Figure 5b, the free carrier density slightly decreases for both the samples annealed in vacuum and $O_2$ at 300 K, although acceptor-like $V_{Ga}+V_O$ and $O_i$ defects are simultaneously passivated by annealing. This reduction in carrier density might be caused by the increase in $V_{Ga}$ originating from the recovery of $V_{Ga}+V_O$ and $O_i$ in bulk $Ga_2O_3$ [10]. The $V_{Ga}$ can act as electron traps to capture electrons, thus pushing the Fermi level away from the conduction band to reduce the carrier density [10,37–39].

**Table 1.** The carrier density and activation energy of Si-doped $Ga_2O_3$.

| Si-Doped $Ga_2O_3$ | Carrier Density (E17) | Activation Energy (meV) | References |
|---|---|---|---|
| film | 13.2 | 10.1 | [13] |
| film | 0.275 | 34.9 | [14] |
| film | 0.1 | 45.6 | [40] |
| film | 7.8 | 33.7–45.6 | [41] |
| bulk | 0.75 | 110 | [17] |
| bulk | ~10 | 15–17 | [12] |
| this work | 4.5–5.0 | 15.6–21.8 | |

Based on the plot of carrier density versus temperature, the thermal activation energies of the carriers for all samples are extracted using the following Arrhenius equation (see, Figure 5b) [42]:

$$N = n_0 \exp(\frac{-E_a}{kT}) \tag{1}$$

where $k$ is the Boltzmann constant ($1.38 \times 10^{-23}$ J/K), $T$ is the temperature, $N$ is the electron density measured at $T$, $E_a$ is the activation energy, and $n_0$ is the pre-exponential factor.

The extracted activation energy shows a 16.8 meV for the pristine sample and increases to 21.8 meV for the vacuum-annealed sample. This is caused by the diffusion of Si impurities toward the surface to generate interstitial Si ($Si_i$) [43], which can behave as donors with an activation energy of 150 meV, as reported by Assil et al. [16], thus increasing the activation energy. On the other hand, although the impurities also diffuse in the oxygen-annealed sample, its activation energy decreases to 15.6 meV because the amount of $V_{Ga}$ in oxygen can be increased to provide more sites for occupation by the diffused Si, producing $Si_{Ga}$ as primary shallow donors to reduce the activation energy [12–14]. Interestingly, with different activation energies, the carrier concentration is approximately at room temperature for both annealed samples because more $V_{Ga}$ are formed in the oxygen-annealed sample, behaving as deep acceptor traps to capture electron carriers.

Furthermore, the curves in Figure 5b for the oxygen- and vacuum-annealed samples intersect at around 300 K, indicating comparable carrier densities. In the region of low temperature, the carrier density of the oxygen-annealed sample is higher, which results from its lower activation energy to excite more carriers, as discussed above. Interestingly, its carrier density is lower than that of the vacuum-annealed sample at high temperatures, which originates from the recovery between $V_{Ga}+V_O$ and $O_i$, as reflected in Figure 4a, where more oxygen interstitials remain in the oxygen ambience.

## 4. Conclusions

In conclusion, the effects of annealing on surface residual impurities (Si and Cr) and intrinsic defects in bulk UID $Ga_2O_3$ were identified and investigated. It was found that the recovery between divacancies $V_{Ga}+V_O$ and interstitials ($O_i$) occured during annealing, which improved the crystal quality. Using Raman and PL measurements, the diffusion of residual impurities toward the surface was observed. Interestingly, an extra red PL peak appeared only for the oxygen-annealed sample, which was attributed to the existence of $Cr^{3+}$ impurities, as depicted by SIMS. Moreover, the carrier mobility was improved by oxygen annealing because it could produce more $V_{Ga}$ near the surface to incorporate the diffused Si and Cr impurities to form substitutions of $Si_{Ga}$ and $Cr_{Ga}$, thereby weakening the impurity scattering. Finally, the carrier density of the oxygen-annealed sample was found to be higher than that of the vacuum-annealed sample at low temperatures because the former has a lower impurity activation energy, whereas the result was the opposite at high temperatures due to the recovery between $V_{Ga}+V_O$ and $O_i$. In short, studying the behaviors of these defects can help us to better understand and regulate the optical and electrical properties of $Ga_2O_3$.

**Supplementary Materials:** The following supporting information can be downloaded at: https://www.mdpi.com/article/10.3390/cryst13071045/s1, Figure S1: The calculated various intrinsic defects and a substitutional No by Ho in [11]; Figure S2: The SEM images of the unannealed, Vac-annealed, and $O_2$-annealed samples; Table S1: The fitted data of mobility [44,45].

**Author Contributions:** Writing—original draft preparation, S.W.; software, Z.L.; formal analysis, H.Y.; writing—review and editing, Y.W. All authors have read and agreed to the published version of the manuscript.

**Funding:** This study was supported by the National Natural Science Foundation of China (grant number 62101044).

**Data Availability Statement:** The data that support the findings of this study are available from the corresponding author upon reasonable request.

**Conflicts of Interest:** The authors declare no conflict of interest.

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
