# Peer review of "Effects of Annealing on Surface Residual Impurities and Intrinsic Defects of β-Ga2O3"

_crystals, doi:10.3390/cryst13071045_

Round 1

Reviewer 1 Report

The authors have annealed commercially supplied B-Ga2O3 crystals using two different conditions, vacuum and O2, both at 1000C, and compared the results to an as-received (unannealed sample)

Unlike the authors claim, this is not a systematic study, as only 3 samples have been used, but the results indicate that annealing, especially in oxygen atmosphere changes the Ga2O3 by inducing UID impurity transport towards the surface. This is by no means a new discovery.

However, despite the lack of novelty, the paper is fairly well written and the experimental content is scientifically sound enough to be publised in MDPI Crystals.

Therefore I can recommend the editor to accept the manuscript without a revision.

Author Response

Comments from Reviewer:

The authors have annealed commercially supplied B-Ga2O3 crystals using two different conditions, vacuum and O2, both at 1000 oC, and compared the results to an as-received (unannealed sample)

Unlike the authors claim, this is not a systematic study, as only 3 samples have been used, but the results indicate that annealing, especially in oxygen atmosphere changes the Ga2O3 by inducing UID impurity transport towards the surface. This is by no means a new discovery.

However, despite the lack of novelty, the paper is fairly well written and the experimental content is scientifically sound enough to be published in MDPI Crystals.

Therefore, I can recommend the editor to accept the manuscript without a revision.

Our Response: The co-authors thank the reviewer for the favorable recommendation and additional works will be further conducted according to the suggestions.

Reviewer 2 Report

The revised paper presents an interesting study of the impact of annealing conditions of Ga2O3 substrates on surface residual impurities (Si and Cr) and intrinsic defects. The paper is well-constructed, and the scientific part of this work is supported by obtained results due to the detailed structural characterization. The paper can be accepted after a minor revision and address some comments, as below.

1. The introduction should be extended. The investigated material is commonly studied due to several reasons that were addressed by the Authors but in a not sufficient way. Gallium oxide is an attractive material from different aspects and for different applications. The 'Introduction' should cover those aspects more broadly to give readers general knowledge about the substrates examined in this study.

2. What is the origin of impurities (Si and Cr) in crystal bulk? Is it a common effect for GaO crystals or fabrication-dependent? Please give a broader knowledge about this issue. Maybe those impurities were present as the quality of substrates was not so fine...?

3. Why did the Authors choose the annealing only in oxygen and vacuum ambient? It should be clearly motivated in the text. The processing sequence using GaO substrate includes several different ambiances for particular reasons. It seems that the Autors chosen only oxygen, but why?

4. The Authors mentioned the characterization of nine (9) samples, but it is not observed during the reading of the manuscript. It seems that only two annealed substrates plus one reference were investigated. Please, give some insights into this issue and explain the experimental part of this study.

5. The first sentence in the 'Conclusion' should contain information on what material is under investigation. Otherwise, the sentence is not completed.

none

Author Response

Comments from Reviewer:

The revised paper presents an interesting study of the impact of annealing conditions of Ga2O3 substrates on surface residual impurities (Si and Cr) and intrinsic defects. The paper is well-constructed, and the scientific part of this work is supported by obtained results due to the detailed structural characterization. The paper can be accepted after a minor revision and address some comments, as below.

(1) Reviewer Wrote: The introduction should be extended. The investigated material is commonly studied due to several reasons that were addressed by the Authors but in a not sufficient way. Gallium oxide is an attractive material from different aspects and for different applications. The 'Introduction' should cover those aspects more broadly to give readers general knowledge about the substrates examined in this study.

Our Response: Please see the extended introduction at Line 29-38 on Page 2.

(2) Reviewer Wrote: What is the origin of impurities (Si and Cr) in crystal bulk? Is it a common effect for Ga2O3 crystals or fabrication-dependent? Please give a broader knowledge about this issue. Maybe those impurities were present as the quality of substrates was not so fine...?

Our Response: The impurities (Si and Cr) are common in the Ga2O3 source, purity of which cannot be as high as silicon at present stage. Moreover, impurities can be also unintentionally induced during the fabrication process, for example, from the precursor or the ambience to produce the namely UID Ga2O3. As a result, these impurities are considered extrinsic residuals, which should be not significant for the substrate quality. (See Line 54-55 on Page 2)

(3) Reviewer Wrote: Why did the Authors choose the annealing only in oxygen and vacuum ambient? It should be clearly motivated in the text. The processing sequence using Ga2O3 substrate includes several different ambiances for particular reasons. It seems that the Authors chosen only oxygen, but why?

Our Response: The aim of this work is to investigate the effect of annealing on diverse defects and behaviors in Ga2O3, among which the defect related to oxygen vacancies is the most common in metal-oxide materials. As a result, simple annealing in oxygen is chosen in this work, and the vacuum ambience is also selected for comparison to reflect the effects of oxygen, which is accompanied by another reference sample annealed in both ambiences. (See Line 63-67 on Page 3)

(4) Reviewer Wrote: The authors mentioned the characterization of nine (9) samples, but it is not observed during the reading of the manuscript. It seems that only two annealed substrates plus one reference were investigated. Please, give some insights into this issue and explain the experimental part of this study.

Our Response: The paragraph has been revised to remove the misunderstanding (See Line 107-108 on Page 4). Three samples for unannealed, vac-annealed, O2-annealed were investigated by XRD in this work, which measurement was conducted three times on each sample to remove possible measurement fluctuation.

(5) Reviewer Wrote: The first sentence in the 'Conclusion' should contain information on what material is under investigation. Otherwise, the sentence is not completed.

Our Response: The sentence has been revised by adding “Ga2O3”. (See Line 236-237 on Page 10).

Reviewer 3 Report

The article is devoted to the study of the effect of high-temperature heat treatment on surface residual impurities and intrinsic defects of unintentionally doped (UID) β-Ga2O3. In general, the presented results have a certain novelty and practical significance, and the article itself is written in an accessible and competent language. This work corresponds to the subject of the declared journal and can be accepted for publication after the authors answer a number of questions that the reviewer had during its analysis.

1. The authors should give an abstract explanation of exactly how the defects were annealed, at what temperatures and conditions, in view of the fact that they undergo migration of oxygen vacancies, the mobility of which depends very much on the conditions of external influences.

2. According to the presented data on changes in the intensities of diffraction reflections, a pattern of changes in the texture of the samples is clearly visible, in view of which the authors are invited to calculate the texture coefficients for all the samples under study and plot their dynamics.

3. The profiles and widths of the diffraction lines of the samples subjected to annealing should also be evaluated, since their change will indicate a decrease in distorting deformation factors.

4. The article should consider the possible thermal broadening of the crystal lattice as one of the factors affecting the change in the properties of the material.

5. In describing the results obtained, the authors should make a comparison with other similar results in order to assess their contribution to this direction.

Reviewer 4 Report

  1. 1. In Fig. 3b and c, the peak values of Si and Cr concentrations are shown to be higher in the Vac-annealed case. However, Fig. 1c shows a lower FWHM value for the Vac-annealed sample. The author previously mentioned that the FWHM of the O2-annealed sample is similar to the control sample due to the diffusion of Si and Cr impurities toward the surface. In that case, why does the Vac-annealed sample have a lower FWHM value?

    2. The authors employed vacuum or O2 ambient for device annealing. Considering that typical annealing for semiconductor devices is performed using hydrogen or deuterium gas, are there any specific reasons why the authors chose these particular annealing conditions?

    3. Several papers have discussed the elimination of contaminants and the improvement of device output performance in Ga2O3 using localized heat treatment. Therefore, it would be beneficial to include the following references for readers' better understanding:

    - H. Bae, K.-S. Lee, P. D. Ye, and J.-Y. Park, "Current Annealing to Improve Drain Output Performance of β-Ga2O3 Field-Effect Transistor," Solid-State Electron., 2021.

    4. Please add a SEM or TEM image of the samples to illustrate the device structure in detail.

Author Response

Comments from Reviewer:

(1) Reviewer Wrote: In Fig. 3b and c, the peak values of Si and Cr concentrations are shown to be higher in the Vac-annealed case. However, Fig. 1c shows a lower FWHM value for the Vac-annealed sample. The author previously mentioned that the FWHM of the O2-annealed sample is similar to the control sample due to the diffusion of Si and Cr impurities toward the surface. In that case, why does the Vac-annealed sample have a lower FWHM value?

Our Response: Although the peak value of impurities of Vac-annealed sample is higher, it is near the surface and decreases sharply in the depth-dependent measurement. Relatively representing impurities content, its integration value of the depth-dependent intensity is much lower than the O2-annealed sample, which is consistent with a lower FWHM value for higher crystalline quality. According to the SIMS measurement, normalized concentration of impurities has been relatively extracted for all the samples, in which the relative values of Cr+, Si+, and Ga+ are 1/176/385987, 1/80/2936, and 1/303/31665 for the unannealed, O2-annealed, and Vac-annealed samples, respectively. (See Fig. 3 and Line 140-142 on Page 6)

(2) Reviewer Wrote: The authors employed vacuum or O2 ambient for device annealing. Considering that typical annealing for semiconductor devices is performed using hydrogen or deuterium gas, are there any specific reasons why the authors chose these particular annealing conditions?

Our Response: The aim of this work is to investigate the effect of annealing on diverse defects and behaviors in Ga2O3, among which the defect of oxygen vacancies is the most common in metal-oxide materials. As a result, simple annealing in oxygen is chosen in this work, and the vacuum ambience is also selected for comparison to reflect the effects of oxygen, which is accompanied by another reference sample annealed in both ambiences. (See Line 63-67 on Page 3)

(3) Reviewer Wrote: Several papers have discussed the elimination of contaminants and the improvement of device output performance in Ga2O3 using localized heat treatment. Therefore, it would be beneficial to include the following references for readers' better understanding:

- H. Bae, K.-S. Lee, P. D. Ye, and J.-Y. Park, "Current Annealing to Improve Drain Output Performance of β-Ga2O3 Field-Effect Transistor," Solid-State Electron., 2021.

Our Response: The recommended paper has been added as the reference [21] in this work. (See the Ref [21] on Page 4)

(4) Reviewer Wrote: Please add a SEM or TEM image of the samples to illustrate the device structure in detail.

Our Response:  The SEM images of the surface morphologies of all the three samples are shown in Fig. S2 of the supplementary information, in which degraded surface of O2-annealed sample is observed. (See Fig. S2 on Page 2) Moreover, no functional device was fabricated in this work, which is focused on the annealing effect on the Ga2O3.

Reviewer 5 Report

The manuscript addresses an important issue of the oxide electronic materials – manipulation of structural defects. Gallium oxide is an emerging wide bandgap semiconductor with attractive properties to be used in semiconductor electronics. The authors show that annealing reduces carrier scattering and changes the photoemission spectrum in bulk Ga2O3. Detail analysis of the crystal quality by XRD and Raman spectroscopy have been presented after annealing in vacuum and in oxygen atmosphere. The distribution of defects near the surface and its effect on carrier mobility are clearly demonstrated. The manuscript is well-prepared with 5 figures and 40 references. I can recommend the manuscript for publishing after corrections of misprints.

In Abstract, l. 14-17, the sentences are grammatically incorrect. Please rewrite.

Line 109, “normalized all peaks” is unclear. Please be specific.

Other misprints on l. 38 (“as”), l. 47 (“doner” -> donors), l. 75 (“diameter of (the) laser” spot).

Please correct – “ambiance” through the text. It should be “ambience” or “atmosphere”.

In Abstract, l. 14-17, the sentences are grammatically incorrect. 

There are word misprints.

Author Response

Comments from Reviewer:

The manuscript addresses an important issue of the oxide electronic materials – manipulation of structural defects. Gallium oxide is an emerging wide bandgap semiconductor with attractive properties to be used in semiconductor electronics. The authors show that annealing reduces carrier scattering and changes the photoemission spectrum in bulk Ga2O3. Detail analysis of the crystal quality by XRD and Raman spectroscopy have been presented after annealing in vacuum and in oxygen atmosphere. The distribution of defects near the surface and its effect on carrier mobility are clearly demonstrated. The manuscript is well-prepared with 5 figures and 40 references. I can recommend the manuscript for publishing after corrections of misprints.

(1) Reviewer Wrote: In Abstract, l. 14-17, the sentences are grammatically incorrect. Please rewrite.

Our Response: The abstract has been revised. (See Line 14-19 on Page 1)

(2) Reviewer Wrote: Line 109, “normalized all peaks” is unclear. Please be specific.

Our Response: The definition of “normalized all peaks” has been added. (See Line 124 on Page 5)

(3) Reviewer Wrote: Other misprints on l. 38 (“as”), l. 47 (“doner” -> donors), l. 75 (“diameter of (the) laser” spot).

Our Response: All the parts have been revised. (See Line 51 on Page 2,Line 59 on Page 3, and Line 88 on Page 4)

(4) Reviewer Wrote: Please correct – “ambiance” through the text. It should be “ambience” or “atmosphere”.

Our Response: All the word has been revised. (See Line 60 on Page 3, Line 78 on Page 3, Line 81 on Page 3, Line 128 on Page 5, Line 178 on Page 8, Line 182 on Page 8, Line 223 on Page 10, and Line 234 on Page 10)

Round 2

Reviewer 3 Report

The authors have answered all the questions, the article can be accepted for publication.

Reviewer 4 Report

All of the comments were reflected in the revised manuscript. I recommend publishing this paper as is.